# Alzheimer’s Amyloid Hypothesis and Antibody Therapy: Melting Glaciers?

**DOI:** 10.3390/ijms25073892

**Published:** 2024-03-31

**Authors:** Poul F. Høilund-Carlsen, Abass Alavi, Rudolph J. Castellani, Rachael L. Neve, George Perry, Mona-Elisabeth Revheim, Jorge R. Barrio

**Affiliations:** 1Department of Nuclear Medicine, Odense University Hospital, 5000 Odense C, Denmark; 2Research Unit of Clinical Physiology and Nuclear Medicine, Department of Clinical Research, University of Southern Denmark, 5230 Odense M, Denmark; 3Department of Radiology, Perelman School of Medicine, University of Pennsylvania, Philadelphia, PA 19104, USA; abass.alavi@pennmedicine.upenn.edu; 4Department of Pathology, Feinberg School of Medicine, Northwestern University, Chicago, IL 60611, USA; rudolph.castellani@nm.org; 5Gene Delivery Technology Core, Massachusetts General Hospital, Boston, MA 02114, USA; rneve@mit.edu; 6Department of Neuroscience, Developmental and Regenerative Biology and Genetics of Neurodegeneration, University of Texas at San Antonio, San Antonio, TX 78249, USA; perry2500@gmail.com; 7The Intervention Centre, Division of Technology and Innovation, Oslo University Hospital, 0372 Oslo, Norway; monar@ous-hf.no; 8Institute of Clinical Medicine, University of Oslo, 0313 Oslo, Norway; 9Department of Molecular and Medical Pharmacology, David Geffen UCLA School of Medicine, Los Angeles, LA 90095, USA

**Keywords:** Alzheimer’s disease, amyloid-PET, ARIA, ATN, CDR-SB, MICD, MMSE, RCT

## Abstract

The amyloid cascade hypothesis for Alzheimer’s disease is still alive, although heavily challenged. Effective anti-amyloid immunotherapy would confirm the hypothesis’ claim that the protein amyloid-beta is the cause of the disease. Two antibodies, aducanumab and lecanemab, have been approved by the U.S. Food and Drug Administration, while a third, donanemab, is under review. The main argument for the FDA approvals is a presumed therapy-induced removal of cerebral amyloid deposits. Lecanemab and donanemab are also thought to cause some statistical delay in the determination of cognitive decline. However, clinical efficacy that is less than with conventional treatment, selection of amyloid-positive trial patients with non-specific amyloid-PET imaging, and uncertain therapy-induced removal of cerebral amyloids in clinical trials cast doubt on this anti-Alzheimer’s antibody therapy and hence on the amyloid hypothesis, calling for a more thorough investigation of the negative impact of this type of therapy on the brain.

## 1. Introduction

The amyloid cascade hypothesis, proposed over 30 years ago, has not yet been proven, and testing of anti-amyloid monoclonal antibodies for anti-Alzheimer’s disease (AD) therapy for 15 years has failed to yield convincing results. The two are linked because in the absence of evidence for the former (stating that AD is caused by cortical deposition of amyloid plaques), an effective hypothesis-based treatment would be the best evidence of causality.

We believe that both the hypothesis and the hypothesis-based immunotherapy suffer from a lack of proper validation and misinterpretations, which we will outline below. We are in favor of Bertrand Russell’s suggestion that whoever postulates that a teapot, too small to be seen by telescopes, orbits the Sun somewhere in space between the Earth and Mars, has the burden of proof, rather than shifting the burden of disproof to others. This is what has happened with the amyloid hypothesis, which strangely is still alive despite a large body of evidence that challenges it [1,2,3,4,5].

The pharma companies that manufacture amyloid-positron emission tomography (PET) tracers or anti-amyloid monoclonal antibodies for passive anti-AD therapy still believe in the amyloid hypothesis, as evidenced by the billions of dollars spent on basic and clinical antibody trials. Indeed, the Food and Drug Administration (FDA) has approved two such antibodies, aducanumab and lecanemab, based on a presumed removal of cerebral amyloids and, in the case of lecanemab, also an uncertain clinical effect [6,7]. A third antibody, donanemab, claimed by pharma to have some clinical effect [8], is under FDA consideration.

Against a backdrop of over 30 years of the amyloid hypothesis and over 20 years of attempts to treat AD based on the hypothesis, one would think that there has been a breakthrough in terms of understanding, diagnosis, and treatment of the disease. In the following, we will take a closer look at this “breakthrough”, with a particular focus on passive immunotherapy of AD.

## 2. Disease Understanding and Diagnosis

The literature is full of tentative and speculative explanations for the pathophysiological associations in AD, one of which is shown in Figure 1.

According to *Britannica*, disease is *“any harmful deviation from the normal structural or functional state of an organism, generally associated with certain signs and symptoms and differing in nature from physical injury. A diseased organism commonly exhibits signs or symptoms indicative of its abnormal state. Thus, the normal condition of an organism must be understood in order to recognize the hallmarks of disease. Nevertheless, a sharp demarcation between disease and health is not always apparent”*.

*Britannica* continues: *“The study of disease is called pathology. It involves the determination of the cause (etiology) of the disease, the understanding of the mechanisms of its development (pathogenesis), the structural changes associated with the disease process (morphological changes), and the functional consequences of those changes. Correctly identifying the cause of a disease is necessary to identifying the proper course of treatment”* [10].

The description and its caveats fit the AD disease area to a worrying degree. Approximately 120 years after Alois Alzheimer described the symptomatology of Frau Auguste Deter [11,12], we still have much to learn about the normal function of the brain. It is therefore not surprising that we still know little about the real causes of AD. It does not help that poorly understood and insufficiently validated markers such as amyloids and tau are given more weight than both clinical judgment and structural and functional image-based examination as conducted in the 2018 diagnostic ATN classification, in which “A”, “T”, and “N” stand for “amyloid”, “tau”, and “neuropathology”, respectively [13]. We still do not know the cause of the disease, and there is no silver bullet that can tell with reasonable certainty whether a patient has AD or not [14,15]. Nonetheless, the upcoming revision of the ATN classification actually proposes an “AT classification” by suggesting that the diagnosis is based solely on A and T, while both clinical assessment and imaging are rejected [16]. 

A look at one of multiple recent studies comparing the interrelationship between A and T and clinical assessment shows—opposite to the authors’ own assurance—not a single relationship between results of amyloid-PET, tau-PET, and clinical judgment that is even remotely close enough to be of clinical relevance in individual patients. Instead, the many plots in the paper show multiple hailstorms in which it is impossible to insert a straight or curved line to indicate a trend (Figure 2) [17]. There is uncertainty even in the pathological field, since a causal relationship between so-called hallmark lesions and functional consequences has not been demonstrated, which is necessary (in *Britannica*’s wording) “to identifying the proper cause of treatment”.

We tend to give special importance to pathological assessment, even though much brain research suffers from the fact that, with rare exceptions (e.g., certain oncological and acute cases), a neuropathological assessment is by nature almost never available at the time it is most needed for the doctors to choose the most appropriate treatment, since this evaluation is made years later, when the patient has died. 

The literature reports an increasing prevalence of amyloid-PET positivity with age. In one study, values ranged from 10% at age 50 years to 44% at age 90 years in cognitively unimpaired persons [18]. In another, they ranged from 2.7% in cognitive healthy persons aged 50 to 59 years to 41.3% in those aged 80 to 89 years [19]. Amyloid positivity was also reported in up to 93% of cognitively unimpaired elderly, aged 78–94 [20]. Thus, probably 50% of all cognitively unimpaired elderly persons are amyloid-PET-positive when they die. So, amyloid-PET positivity indicates a fifty percent risk of developing AD later in life. And then what? Who needs that information, when nobody knows for sure what the findings signify, when there is no effective treatment, and there are no effective measures to prevent the disease from occurring? Additionally, given that about 50% of Lewy body dementia patients and 25% of Parkinson’s dementia patients are also amyloid-positive [21] and that amyloid positivity decreases with age in patients with probable AD [22], how can anyone suggest that amyloid positivity is a reliable hallmark of AD?

Numerous observations show that amyloid-PET is not suitable for the diagnosing or differential diagnosing of dementia patients including AD patients, mainly due to the non-specificity of the amyloid tracers. They tend to show the same range of accumulation in the cortex of patients with probable AD or mild cognitive impairment (MCI) as in older healthy control subjects (Figure 3). The three sets of box and whiskers say it all: there is a huge overlap among the three groups, making it impossible to diagnose probable AD or MCI with reasonable certainty or to distinguish between the two or separate them from older healthy controls. Moreover, all three groups have a significant number of subjects with similarly low accumulation as in healthy young adults who are apolipoprotein E gene (APOE ε4) noncarriers. In other worbds: a negative amyloid-PET scan cannot exclude the diagnoses “probable AD” or “MCI”. The authors of the study came to a different conclusion, as indicated: “The finding of our analysis confirm the ability of [^18^F]florbetapir-PET standardized uptake value ratios (SUVRs) to characterize amyloid levels in clinically probable AD, MCI, and OHC groups” (where OHC = older healthy control) [23].

Beyond the true brain amyloid levels, the problem resides in the lack of specificity of these amyloid-PET probes, which results in PET images that are difficult to interpret. For example, a consistent lack of accumulation of ^11^C-Pittsburgh compound-B ([^11^C]PiB) has been shown in the medial temporal lobe of AD patients, a brain area known to have amyloid deposits based on neuropathological determinations [24]. Instead, heavy uptake into myelin and areas of myelin damage are seen in the white matter in the absence of amyloid aggregates [25,26]. Moreover, [^11^C]PiB accumulation has been seen in moyamoya disease (caused by stenosis/occlusion in the anterior cerebral circulation), probably because [^11^C]PiB, the mother of all the amyloid-PET probes, happens to be a high-affinity substrate for estrogen sulfotransferase SULT1E1, a very common brain enzyme related to inflammation [27] (Figure 4).

Moyamoya disease can be treated with encephaloduroarteriosynangiosis, a surgical procedure in which a segment of a scalp artery is transferred onto the surface of the brain. Amyloid-PET positivity has also been found in the brain after the removal of brain tumors [28], and recently, a case story reports negative [^11^C]PiB-PET findings in a patient with autosomal dominant AD [29]. Therefore, amyloid-PET can neither diagnose AD nor exclude patients without AD and hence should be deemed unsuited for selecting patients for anti-AD trials. These observations were not taken into account when the proposal for revision of the ATN classification maintained that amyloid positivity by amyloid-PET is not only necessary but also sufficient to diagnose AD [16].

Apart from this, all previous attempts to delineate the disease or the syndrome AD fail to point to a possible triggering factor, if there is one. However, it seems unlikely that a progressive and, so far, irreversible cognitive/functional decline would suddenly start all by itself. It is therefore not surprising that inflammation/infection [30,31,32], type 3 diabetes [33], hypotension/hypoperfusion [34,35,36,37,38], and many other possibilities have been implicated [39,40,41]. So, it is to be hoped that in the future more resources will be devoted to these and other possibilities as they may be more effectively prevented and, if so, would represent a real breakthrough. 

## 3. Treatment

A myriad of treatments targeting AD have been tried or are in the pipeline [41,42,43,44,45], as illustrated in Figure 5 from reference [34]. Common to most of these is a belief in the amyloid cascade hypothesis, i.e., that the main principle is the prevention of amyloid deposition (vaccination, gamma-secretase blockers, and β-site amyloid precursor protein-cleaving enzyme 1 (BACE1) inhibitors) or removal of amyloid deposits (passive immunotherapy with monoclonal antibodies). However, a wider range of principles, too numerous to summarize, has also been investigated. 

Here, we concentrate on immunotherapy, distinguishing between active immunotherapy with vaccines and passive immunotherapy with monoclonal antibodies directed against the protein amyloid-beta (Aβ) and/or its oligomers [46]. Active therapy was abandoned in 2005 because 6% of those treated developed meningoencephalitis [47]. The degree of plaque removal after vaccination varied considerably but was quite significant according to the few post mortem assessments that were performed, and yet the patients did not show cognitive improvement. Instead, they developed severe end-stage dementia before death [48,49], a fact seemingly ignored by today’s passive immunotherapy trials that claim to cause removal of cerebral amyloid deposits.

A wide range of monoclonal antibodies targeting cortical amyloid has been tested, but the hoped-for positive results have not materialized. Biogen is halting development of aducanumab [50] in favor of lecanemab, even though it received an accelerated FDA approval 2½ years ago based solely on the supposed removal of amyloids from the brain, with a note that it was “expected to lead to a reduction in the clinical decline of this devastating form of dementia” [6]. Lecanemab obtained accelerated FDA approval in January 2023 and full approval in July 2023 and is claimed to also offer some statistical clinical improvement in terms of delay in cognitive decline. However, major question marks can be placed on the results of the randomized clinical trials (RCTs) that are the basis for the approvals. They primarily concern clinical efficacy, amyloid positivity according to amyloid-PET, and claimed therapy-induced removal of cerebral amyloids.

### 3.1. Clinical Efficacy

A major question concerns the fact that reported clinical effects do not reflect improvements in the patients’ disease states but instead reflect only a very minor delay in cognitive/functional decline. In the case of aducanumab, no definite clinical effect was claimed, presumably because two large, almost identical phase 3 RCTs, EMERGE and ENGAGE, had opposing clinical effects that canceled each other out [51].

Regarding lecanemab and donanemab, in addition to presumed amyloid removal, they are claimed to cause a delay in progressive cognitive/functional decline. The respective CLARITY AD [52] and the TRAILBLAZER-ALZ 2 [53] trials were conducted in patients considered to have MCI due to AD or mild AD. All patients were “amyloid-positive” according to amyloid-PET and had average cognitive scores on the Mini-Mental State Examination (MMSE) scale (range 0–30, with lower scores indicating greater impairment) and the Clinical Dementia Rating-Sum of Boxes (CDR-SB) scale (range 0–18, with higher scores indicating greater impairment) of about 25.5 and 3.2 and 22.5 and 3.8, respectively, indicating slightly more severely affected patients in TRAILBLAZER-ALZ 2. 

Press releases from pharma have touted the great benefits of the two antibodies in terms of 27% and 36% delays in cognitive decline [54,55], but in reality, these are minimal differences from 1½ years of treatment compared to a placebo. In the lecanemab trial by van Dyck et al., the reported difference was 0.45 points on their primary endpoint, the CDR-SB scale, corresponding to 2.5 percent of the scale range [52] (Figure 6). In the donanemab trial by Sims al., the difference was 3.25 points on their primary endpoint measure, the integrated Alzheimer Disease Rating Scale (iADRS) (range 0–144, with lower scores indicating greater impairment), or 2.3 percent of the scale range, and 0.67 points on the CDR-SB scale equal to 3.7 percent of the scale range [53]. 

These differences in CDR-SB score compared with a placebo are less than the difference shown already by Rountree et al. in 2009 in more impaired patients (CDR-SB 7) treated maximally with conventional AD therapy (cholinesterase inhibitors donepezil, rivastigmine, and galantamine and/or memantine). They found a difference of 0.9 points (5%) in CDR-SB score after 1.5 years of treatment [56]. On the MMSE scale, Sims et al. also reported a difference in favor of donanemab of 0.48–0.57 points or 1.6–1.9% of the scale range [52]. In comparison, a difference of 1.5 points or 5% of this scale has been observed in more diseased AD patients undergoing maximal conventional therapy [57,58]. 

A recent review on the effect of cholinergic therapy for AD showed effects of the same magnitude, providing evidence for the long-term effect of cholinesterase inhibitor therapy suggesting a disease-modifying effect of these drugs [59]. This is consistent with a Swedish study of 11,652 cholinesterase inhibitor users and 5926 non-users followed for 5 years, where the users, measured on the MMSE scale, had a modest reduction in cognitive decline that persisted over the long term (3–5 years). The effect was greatest with galantamine and was associated with a lower risk of death [60].

**Figure 6 ijms-25-03892-f006:**
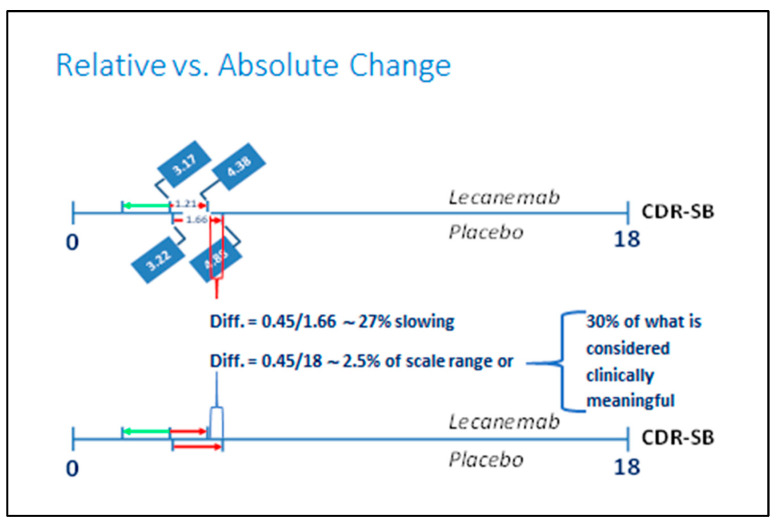
The “27% greater slowing” in cognitive decline with lecanemab as measured on the CDR-SB scale equals only 2.5% of the scale range, corresponding to about ⅓ of what is considered the minimal clinically important difference, according to Andrews et al. [61]. The figure is based on data from reference [52]. Note: the whole length of the CDR-SB scale is presented to indicate the actual minimal size of the 27% “positive” slowing effect touted in lecanemab pharma press releases. The green arrows indicate the change in direction on the scale that would be seen if the treatment caused an actual improvement. In reality, there is only a worsening (indicated by the red arrows) also during lecanemab treatment; it just happens a little slower in the treated patients compared to the placebo patients.

The difference observed with lecanemab on the CDR-SB is less than the 1–2 points that Andrews at al. have suggested as the “minimally important clinical difference” (MICD) on this scale [61]—a view that has been disputed by van Dyck et al. [62] but defended by Andrew et al. [63] with reference to Lansdall et al., who based their findings on the 12-month treatment time point in the Memory Impairment Study of patients with MCI. This study suggested that increases (impairment) of 1–2.5 points on the CDR-SB scale (and of 2–5 points on the 11-item and 13-item Alzheimer’s Disease Assessment Scale-Cognitive (ADAS-Cog) subscales) reflect minimal to moderate levels of deterioration [64], findings recently confirmed by Ebel et al. [65]. Regardless, most clinicians with a modicum of experience in AD and dementia and the fluctuations that characterize these disorders will know that such small differences cannot be diagnosed with reasonable certainty in individual patients. The effects of conventional treatment as measured on the iARDS scale used in the donanemab trial have, to our knowledge, not been reported.

In this light, it is remarkable that the RCTs with these antibodies were not designed to be compared with the traditional standard anti-AD therapy but only with a placebo, the type and potential effects of which were not sufficiently described in the trial reports. It appears that the trials did not account for any concomitant cholinesterase/memantine treatment during antibody treatment, as would normally be performed in drug trials, or for the use of other concomitant medications that may affect incidence or progression of AD, including over-the-counter non-steroidal anti-inflammatory drugs [66,67]. Thus, there is no evidence to suggest that these anti-amyloid treatments have a particularly beneficial effect in patients with early AD, let alone anything that can outweigh the so-called “amyloid-related imaging abnormalities” (ARIAs) [68] and the accelerated brain volume loss [69] that they induce (see below). It is therefore surprising that no long-term follow-up results of trial patients have been published that would indicate what happens when the treatment stops and what the long-term consequences of these threatening side effects are.

### 3.2. Approved Amyloid-PET Imaging Probes

Three compounds have been approved by the FDA for amyloid-PET imaging: [^18^F]florbetaben (Neuraceq^®^), [^18^F]florbetapir (Amyvid^®^), and [^18^F]flutemetamol (Vizamyl^®^), and recently, the Centers for Medicare & Medicaid Services (CMS) extended coverage of amyloid-PET scans in patients with dementia in the context of clinical studies [70]—a dangerous path that could lead to unnecessary amyloid-PET scans and also make non-AD patients candidates for a treatment with serious side effects that does not work better than conventional cholinesterase inhibitor/memantine treatment [71] (see below).

Amyloid-PET imaging probes are indicated for the assessment of brain amyloid content and have even been repeatedly proposed in the literature as a diagnostic tool for AD. On this basis, amyloid-PET imaging has been utilized for the selection of subjects to participate in monoclonal anti-amyloid therapies and to assess their efficacy by measuring brain amyloid-PET imaging signals pre- and post-treatments. However, a host of fundamental limitations, consistently exposed in the literature, question assessments of the validation of amyloid-PET probes for the intended purposes [23]. Of particular concern are the lack of support for the in vivo specificity of these PET probes, recognized early by Villemagne et al. [72], the target specificity for tertiary forms of Aβ as shown in autopsy, and the extensive uptake of amyloid-PET probes in the white matter [24,25], whose intensity is consistently superior to the signal intensity in the grey matter layer in AD, as described in detail earlier [23]. Tracer retention in the white matter for all FDA-approved PET imaging probes is in fact also markedly higher than that in the grey matter in all cases [23]. This is consistent with a more complex pattern of accumulation than attributable to non-specific retention due to the hydrophobic properties of the tracers. Moreover, this high white matter retention complicates reading of grey matter PET signals in Alzheimer’s disease due to partial volume and signal spillover effects, which constitute an additional impediment to appropriate quantification of the PET signal to interpret the effect of the monoclonal anti-amyloid therapies [73,74]. In addition, using [^11^C]PiB, Klunk et al. acknowledged earlier the non-specific binding of existing Aβ tracers limiting the sensitivity of PET imaging, especially in a prodromal phase of AD when plaque levels might be low [75], which is the situation encountered when asymptomatic subjects or subjects with early symptoms are hired for anti-amyloid clinical trials.

The discovery that [^11^C]PiB and [^18^F]flutemetamol, uncharged derivatives of the histological dye Thioflavin-T (ThT) (Figure 7), are high-affinity substrates for estrogen sulfotransferase SULT1E1 [76], an enzyme found in rodent and human brains, demonstrates a source of non-specificity of these two ligands and may explain why they have been shown to produce very similar patterns in vivo with PET [77]. Upon reaction with the enzyme, their 6-O-substrate is produced in vivo and can be retained in tissue, similar to the formation of fluorodeoxyglucose (FDG)-6-phosphate upon reaction with tissue hexokinase. The in vivo reactivity of [^11^C]PiB with estrogen sulfotransferase SULT1E1 in humans can be confirmed as its main metabolite in plasma, resulting from peripheral metabolism of its 6-O-sulphate. Moreover, the demonstration that using [^11^C]PiB estrogen sulfotransferase detection has been recently observed for the first time in vivo in humans lacking amyloid brain deposition [26] confirms the need to demonstrate that in vivo PET signals obtained with “amyloid-specific” imaging PET probes are not the result of estrogen sulfotransferase of other possible tissue targets.

The other two approved amyloid-PET imaging probes ([^18^F]florbetapir and [^18^F]florbetaben) are members of the trans-stilbene family. As with the observations with the ThT amyloid-PET probes, whereas the in vitro evidence of amyloid specificity is persistent in the literature, in vivo specificity of these probes for amyloid deposition is lacking. In vitro experimental conditions are quite different from in vivo settings, where probe metabolism, transport, and off-target enzyme tissue are elements that would influence and drastically change the interpretation of in vivo results. For example, [^18^F]florbetapir has been reported to be rapidly metabolized peripherally in rodents with a very significant amount of metabolites (44%) present in the brain at 2 min after injection. This resulted from the blood–brain barrier (BBB) brain penetrability of peripheral metabolites, as confirmed by Avid Pharmaceuticals investigators [78], with similar results expected in humans. With such a high level of brain metabolites and concomitant depletion of the parent probe, added to the in vivo radiodefluorination also reported for both [^18^F]florbetapir and the structurally related [^18^F]florbetaben [79], the possibility of accurate measurement of amyloid loads at the time of PET image analysis is virtually eliminated [23], even accounting for the expected metabolism translation from rodents to primates that would favor slower peripheral metabolism in humans. The rapidly increasing concentrations of brain metabolites of [^18^F]florbetapir and [^18^F]florbetaben produce a high background signal and hinder signal specificity for the amyloid deposition observable in in vitro experiments. Based on the sensitivity tracer kinetic models applicable for quantification, only limited BBB transport of metabolites would be acceptable for quantification, but the magnitude of the reported peripheral metabolism of [^18^F]florbetapir and [^18^F]florbetaben makes it very unlikely that this would be the case. Brain transport of peripheral metabolites through the BBB is one of the most basic considerations in the design and utilization of optimal PET imaging probes [23,80].

An additional question with these approved amyloid-PET probes is the binding selectivity for the tertiary forms of amyloid (i.e., hard core, diffuse plaques) and how well the amyloid-PET signals quantitatively correlate with autopsy findings. On that front, the bibliography shows contradictory reports on the binding selectivity for tertiary amyloid forms. Whereas it has been claimed that [^11^C]PiB strongly labels fibrillar amyloid and only weakly diffuse plaques [81], others indicate that this differentiation is not possible [82]. Some authors conclude that [^11^C]PiB in vivo PET imaging and post mortem evaluation showed limited overall agreement with the Consortium to Establish a Registry for Alzheimer’s Disease (CERAD) neuritic plaque ratings [83]. 

### 3.3. Amyloid Positivity

A second major question refers to the fact that all RCTs have “amyloid positivity” by amyloid-PET or CSF biomarkers as an entry criterion. With the existing limitations of the current amyloid-PET probes, as indicated above, this is debatable, given two interrelated conditions that cannot be clearly separated. 

One is the insufficient specificity of amyloid-PET tracers, which means they mark inflammation [26,84] and/or myelin [85,86,87] and myelin damage [24,25] in the absence of amyloids, which is a plausible reason why they are avidly taken up in the grey and the white matter, in controls, and in this category of patients (Figure 8 and Figure 9), even though amyloids in the early stages of disease are supposed to be only localized in the former [88,89].

The second is that less than half of pathologic AD cases are not considered to clinically have AD [90]. Amyloid positivity by PET is present in more than 40% of cognitive healthy subjects older than 90 years, as demonstrated by Jansen et al. who in a meta-analysis of 2914 participants with normal cognition and 697 with subjective cognitive impairment and 3972 with MCI aged 18–100 years found an increase in amyloid-PET positivity from 10% at age 50 to 44% at age 90 among subjects with normal cognition [18]. The same group was observed in a meta-analysis of 1359 patients with clinically diagnosed AD, showing a decrease in amyloid positivity by PET from age 50 to 90 years in APOE ε4 non-carriers from 86% to 68% and to a lesser degree in APOE ε4 carriers [22], and a tendency, although less pronounced, was also observed in a study on CSF-based versus amyloid-PET-based estimates of amyloid positivity in cognitive healthy subjects and patients with increasing degrees of AD [91]. Thus, with the uncertainties resulting from the poor specificity of these PET imaging probes, readers of these amyloid-PET scans are left questioning about the proper attribution of the cortical signal.

Many explanations or rationalizations for the amyloid positivity of cognitive healthy subjects have been put forward, including the claim that such individuals will sooner or later develop AD if they live long enough [92]. But certainly, amyloid positivity does not unequivocally mean that this would be the case and exposes the high likelihood that many participants in these anti-amyloid clinical trials would never develop or have AD or even have significant amyloid deposition in their brains.

As indicated earlier, the term “biologically defined” as opposed to “clinically defined” AD has been introduced as a core concept in the ATN classification of the National Institute of Aging-Alzheimer’s Association Research Framework [12] without scientific evidence. The finding of these biomarkers, which are more frequent at all ages and three times more prevalent at age 85+ than clinically proven AD, has been used to argue that clinical assessment can be dispensed with altogether [93]. The upcoming revision of the ATN classification goes one step further by suggesting that imaging of neurodegeneration also be omitted so that the definition of AD is based solely on biomarkers [15], the measurement of which is questionable, the significance of which is unknown, and whose diagnostic validity has never been properly proven. The implication is that amyloid-PET positivity does not represent a specific cerebral disease but rather, at best, a generalized neurodegenerative disorder in large parts of the brain. Since amyloid positivity has been observed in patients with all types of dementia and in controls, it further implies that using amyloid positivity as an entry criterion in all antibody RCTs will ensure the inclusion of “amyloid positive” subjects who do not suffer from AD. 

### 3.4. Amyloid Removal

A third major question refers to the alleged removal of cerebral amyloids resulting from monoclonal antibody treatment. The aducanumab, lecanemab, and donanemab trials report up to 27%, 55%, and 85% amyloid removal, respectively [52,53,94], based on an apparent reduction of the amyloid-PET signal, without consideration of additional factors associated with these treatments. Significant brain tissue damage (ARIA H and ARIA E, among others) has been reported as a result of these monoclonal antibody treatments [74,95,96]. Clearly, the reported signal decline in the white matter of AD patients (Figure 10), also shown on images in the Sevigny et al. aducanumab trial report, cannot be attributed to amyloid removal [94].

**Figure 8 ijms-25-03892-f008:**
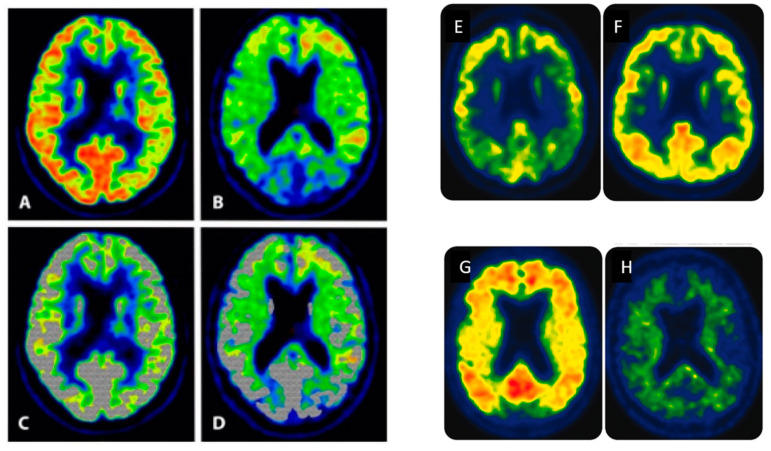
Topographic difference in FDG and [^11^C]PiB uptake in AD patients. The block of four on the left is from reference [96] and shows PET images from a patient in his early 70s with presumed mild AD. The FDG uptake represents glucose metabolism in the grey matter, localized in a narrow band near the skull (**A**), while the [^11^C]PiB uptake extends more centrally (**B**), indicating non-specific uptake in the white matter. This is more evident when the shaded grey area, representing FDG uptake (**C**), is superimposed on the [^11^C]PiB image (**D**). The uptake ratio between grey and white matter was 4–5:1 for FDG and 1:1 for [^11^C]PiB. The block to the right is a modified version of Figure 1 in reference [92]. Two MRI images (**C**,**D**) have been removed and are shown separately in Figure 9. (**E**,**F**) show the FDG uptake in a 75-year-old patient with typical AD and a 71-year-old normal subject, respectively. (**G**) shows [^11^C]PiB uptake in a 71-year-old woman with AD; (**H**) shows [^11^C]PiB uptake in a 93-year-old normal person. Notice similar FDG uptake in the normal subjects ((**A**) in the left block and (**F**) in the right block) found in a narrow peripheral brim, whereas [^11^C]PiB uptake extends into the white matter of the AD patients ((**B**) in the left and (**G**) in the right block). The size ratio between grey and white matter is normally about 3:2 but more likely 2:2 or 2:3 in AD patients, so the majority of the [^11^C]PiB signal in (**G**) in the block on the right probably originates from the white and not the grey matter. The block to the left is reproduced from Høilund-Carlsen, P.F. et al. [96], 2023, an open-access article under the CC BY license. The block to the right is reprinted with permission from Jack et al. [92], 2016, *Neurology*.

**Figure 9 ijms-25-03892-f009:**
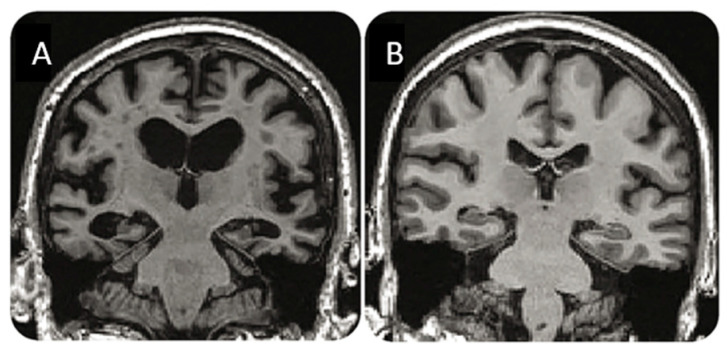
MRI changes in AD. MRI images cut out of Figure 1 in reference [92]. (**A**) shows pronounced cortical atrophy and significant ventricular enlargement in a 71-year-old male AD patient, while (**B**) shows conditions in a 71-year-old healthy female. The thin strip of cortical tissue in (**A**) demonstrates how difficult, if not impossible, it is to segment the cortical areas in AD patients. Reprinted with permission from Jack et al. [92], 2016, *Neurology*.

**Figure 10 ijms-25-03892-f010:**
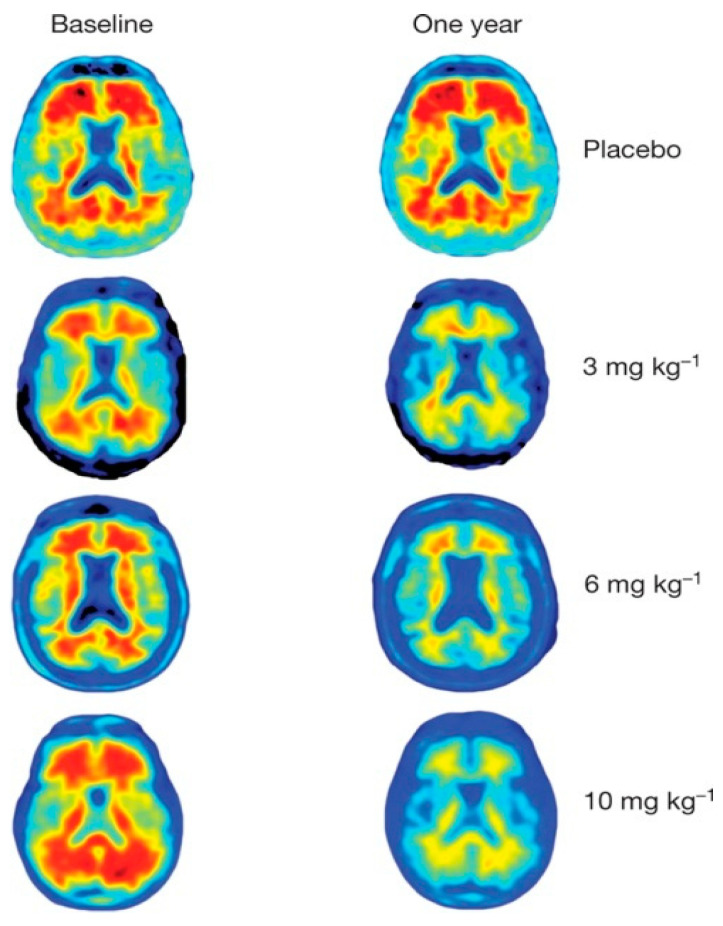
Cerebral amyloid-PET images using [^18^F]florbetapir from reference [94]. Images at baseline and after one year from one placebo patient and three patients treated with three different doses of aducanumab. Note the high PET signal from the white matter at baseline in all patients and the very low signal after one year in the treated patients. Note also that the uptake in the cortex is barely visible in any of the patients. Reprinted with permission from Sevigni et al. [94], 2016, Springer Nature.

This does not mean that tracers do not also target amyloids in the grey matter. They may do it when amyloids are part of an antibody–antigen reaction, which, when particularly pronounced, gives rise to ARIAs as a result of focal tissue damage. That the edematous ARIA-E type, unlike the ARIA-H type, characterized by microhemorrhages, tends to gradually “resolve” on MRI during the treatment course is by no means a guarantee that permanent cellular damage has not occurred. The persistence of the ARIA-Hs suggests otherwise. Therefore, it is worrying that these changes and their long-term impact on brain function still have not been properly investigated, especially since this can be easily conducted using functional imaging in the shape of FDG-PET. 

This treatment-induced brain damage makes it difficult to accurately quantify the regional imaging signal, and it is virtually impossible to determine whether cortical decline is solely due to cortical amyloid deposition. The grey matter width of typically less than 3 mm in AD patients (Figure 10) is well below the spatial resolution of 5 mm or more in tomographic acquisitions with modern PET scanners. Moreover, the activity in the grey matter is heavily influenced by spillover from non-specific activity in nearby structures [97], i.e., from the bone marrow of the skull [85,98], and in particular from the intertwined white matter [85,86,87], the contribution from which cannot be eliminated [73,99].

Hence, the claimed “clearance” of amyloids during therapy can hardly be explained as anything other than a decreasing amyloid-PET signal heavily influenced by a generalized antibody–antigen inflammatory process causing significant additional brain damage, as evidenced in autopsy determinations [100,101,102]—an assumption that is also strongly supported by the accelerated loss of global brain volume that characterizes all forms of anti-AD antibody therapy [69].

The severe and widespread antigen–antibody reaction is also evidenced by the few neuropathological examination reports available from patients who have died during antibody therapy [100,101,102]. They show the result of extensive antibody–antigen reactions and significant macrophage invasion. The latter may give an impression of amyloid removal, as amyloids are also seen being ingested by macrophages. However, many more structures, in particular the vessels, are affected. Recruitment of many mononuclear cells is not the brain’s normal response to injury. This is characterized by a relative lack of neutrophils and a delay in monocyte recruitment followed by a rapid transformation of the recruited monocytes into microglia [103].

In immune-mediated inflammation, activated circulating T-cells directed against a brain antigen are abundant, together with activation of additional humoral immune responses including cytokines, i.e., mechanisms of low specificity which mainly induce “bystander” damage of the central nervous system (CNS) tissue including white matter myelin damage [104], a side mechanism that is also evident in non-immune-mediated injury [103] and one of the potential mechanisms that is responsible for the increased non-specific accumulation of amyloid-PET tracers in antibody RCTs. This appears to be the situation in patients who have died after immunotherapy [100,101,102]. The changes resemble those seen after intracerebral hemorrhage, where there is an inflammation-driven breakdown of the BBB with recruitment over days of circulating inflammatory cells and subsequent secondary immunopathology [105].

In light of the above, we find it surprising that antibody therapy has not been linked to more deaths than the relatively few that have been reported thus far [106]. We find it even more remarkable that so few and not all deaths in antibody trials have been subjected to neuropathological examination.

### 3.5. Summing Up

The issues and questions raised in this article are displayed in Table 1. The three major questions, i.e., clinical efficacy of less than that of conventional AD therapy, unspecific amyloid positivity as an entry criterion in the antibody trials, and misinterpretation of a decreasing amyloid-PET signal during treatment, all suggest the conclusion that therapy with the monoclonal antibodies aducanumab, lecanemab, and donanemab provides only little, if any, benefit to patients.

This verdict seriously questions the efficacy and value of these treatments predicated under the domain of the amyloid hypothesis. The perceived therapy-induced removal of cerebral amyloids is more likely to be heavily influenced by a decreasing amyloid-PET signal due to therapy-related cerebral damage hampering the uptake of non-specific amyloid-PET tracers. This is consistent with the increased tendency for ARIAs and the accelerated loss of brain volume that characterizes this kind of therapy and calls for in-depth investigation of the potentially negative impact of passive immunotherapy on the brain.

## 4. Perspectives

In light of the above, we find it impossible to point to new diagnostic and therapeutic options based on the amyloid hypothesis, which despite numerous scientific publications remains unjustified, as demonstrated by the lack of efficacy of immunotherapy with monoclonal antibodies. We cannot rely on therapies that do not result in a clear positive effect in RCTs with nearly 1000 patients in each arm and, as in the lecanemab and donanemab trials [52,53], showing only a marginal statistical delaying effect on cognitive/functional status after weeding out those patients who could not tolerate the treatments, an effect that is even less than with conventional cholinesterase inhibitor/memantine therapy and less than what is considered the MICD on common cognitive scales.

It is high time to turn our attention to new and more promising diagnostic approaches, as well as new preventive and therapeutic options, since the last 20–30 years of amyloid-based efforts have ended in an unproductive stalemate. We hope that the brains of patients who have received treatment with monoclonal antibodies in various RCTs do not, as we fear, suffer further damage. We recommend that the FDA and CMS take action to document and ensure that this is not the case, including requiring in vivo testing with FDG-PET, the only known method that can reveal cerebral functional impairment, as well as mandating autopsy and neuropathological examination of all patients who die in connection with passive immunotherapy of AD.

We have previously highlighted infection and hypoperfusion due to cerebral microcalcifications as some of the triggers that may be addressed, suggesting low-dose radiation as an interesting therapeutic option that has a surprisingly positive effect in certain diseases, including cancer [107]. There are several other approaches that should be considered. However, the most important thing is to shift the focus away from amyloids and towards more promising and rational options so that we do not continue to stall rather than help tragically ill patients who remain without real therapeutic possibilities.

## 5. Conclusions

Alzheimer’s research has reached a stalemate due to an increasingly artificial disease definition and delineation based on non-specific biomarkers and driven by ineffective anti-amyloid treatments. The focal point has been the unproven amyloid hypothesis, ersatz disease definitions like the ATN/AT classification, which no clinician can relate to, non-diagnostic amyloid-PET, and most recently, an alleged statistically defined clinical effect, for example, for lecanemab and donanemab, of minimum quantitative relevance, with adverse effects deserving to be fully investigated and not dismissed before moving forward. Instead, future efforts should be focused on new and more promising options than the fruitless amyloid-based approaches. There is reason to return to methods that are unmistakably related to the nature of the disease, namely clinical assessment and neuronal imaging like FDG-PET, to determine cognitive status and cerebral function.

## Figures and Tables

**Figure 1 ijms-25-03892-f001:**
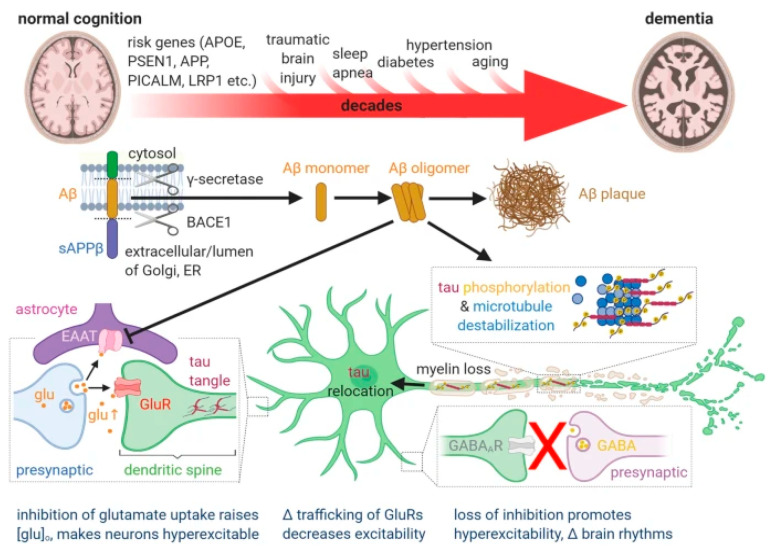
Schematic representation of putative pathophysiological associations in AD from a review article and with the following legend: “Current, generally held ideas about the pathology underlying Alzheimer’s disease (see main text for details). The transition from normal cognition to dementia, over decades, is promoted by the risk factors shown above the large red arrow. Aβ is produced from amyloid precursor protein (APP) by the action of the γ secretase and β secretase (BACE1) as monomers, but these can then form soluble oligomers, which ultimately form extracellular precipitates as amyloid plaques. Aβ oligomers inhibit astrocyte glutamate uptake (EAAT), thus potentiating the action of synaptically released glutamate (glu). This, together with a loss of GABAergic inhibition, leads to some neurons becoming hyperexcitable. Meanwhile, Aβ oligomers also induce hyperphosphorylation of axonal microtubule-associated tau, which leads to tau redistributing partly to dendrites where it disrupts trafficking of glutamate receptors and thus depresses excitation and neuronal firing. These synaptic effects, and Aβ- and/or tau-induced loss of axonal myelin, may induce cognitive dysfunction well before synapses are lost and neurons die. The levels of Aβ oligomers and hyperphosphorylated tau correlate better with cognitive decline than does the level of Aβ plaques.” Reproduced from Korte et al. [9], 2020, an open-access article under the CC BY license.

**Figure 2 ijms-25-03892-f002:**
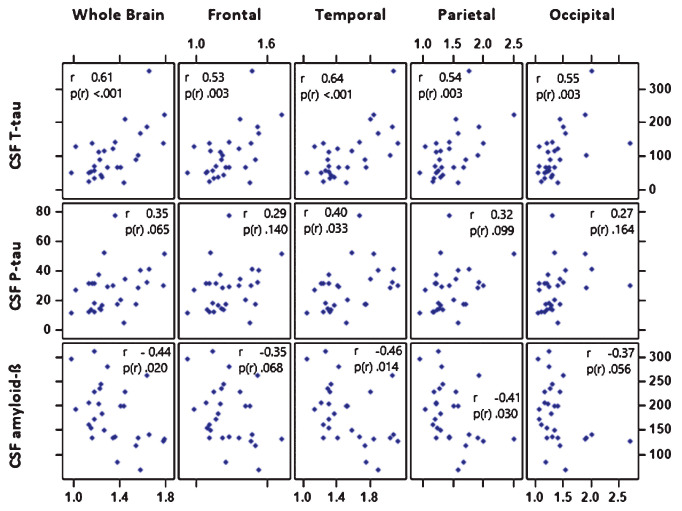
Scatter plots of global and regional [^18^F]flortaucipir PET standardized uptake value ratios (SUVRs) and cerebrospinal fluid (CSF) biomarkers from a study on the associations of [^18^F]flortaucipir with CSF amyloid and CSF tau biomarkers in whole brain and various brain regions. *X*-axis: global and lobar cranial SUVRs for ^18^F-flortaucipir. Inset: r = Pearson’s coefficient (unadjusted); *p*(r) = significance level. T-tau, total tau; P-tau, phosphorylated tau; CSF, cerebrospinal fluid; SUVR, standardized uptake value ratio. CSF biomarker unit in pg/mL. Reprinted with permission from Okafor et al. [17], 2020, *Journal of Alzheimer’s Disease*.

**Figure 3 ijms-25-03892-f003:**
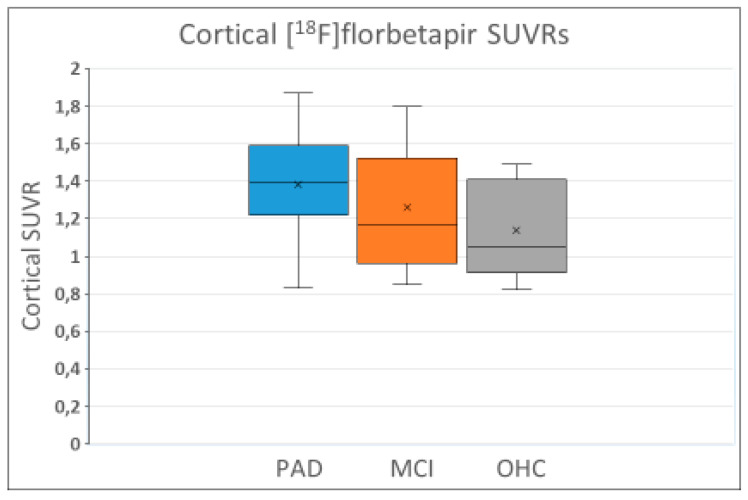
Box and whisker plots constructed based on cortical standardized uptake value ratios (SUVRs) presented for the following clinical diagnostic groups: probable AD (PAD), n = 68, mild cognitive impairment (MCI), n = 60, and older healthy control (OHC) group, n = 82, in Fleisher et al. [23], 2011, *Archives of Neurology*. Crosses in the middle of the boxes are mean values, black horizontal lines represent medians. Note the huge overlaps between groups and that all groups include subjects with values ≤ 1.08 that according to Fleisher et al. are the upper limit for cortical SUVRs in young adult APOE ε4 noncarriers. What Fleisher et al. term “global mean cortical SUVRs” are the average of the mean values registered in the following cortical regions of increased [^18^F]florbetapir signal: medial orbital frontal, temporal, anterior, and posterior cingulate; parietal lobe; and precuneus.

**Figure 4 ijms-25-03892-f004:**
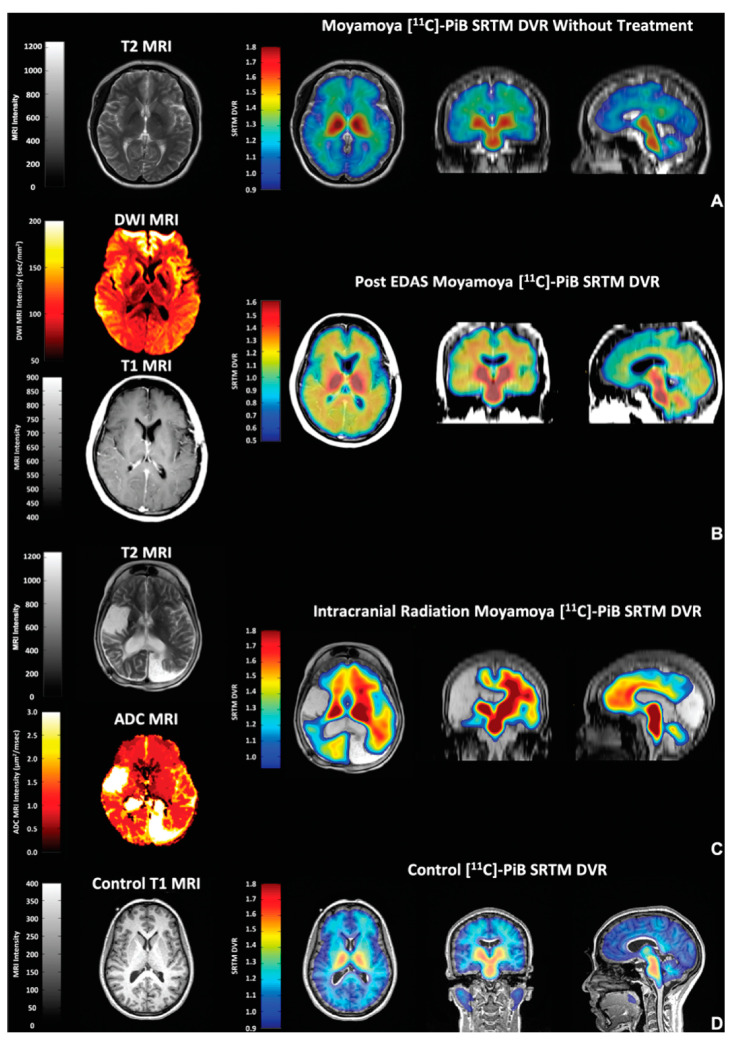
Chronic inflammatory signal from [^11^C]PiB moyamoya syndrome. The figure displays the heterogeneous [^11^C]PiB PET response in various moyamoya cohort subjects lacking brain amyloid deposition: (**A**) with no surgical intervention of any kind; (**B**) after receiving an EDAS procedure for cerebral revascularization demonstrating a sustained subcortical [^11^C]PiB uptake; (**C**) a patient with moyamoya syndrome with a ventriculoperitoneal (VP) shunt and severe trauma from sequelae of radiation treatment for childhood astrocytoma, as featured in the T2 and ADC MRIs; (**D**) a control subject with a normal T1 MRI and a [^11^C]PiB PET normal profile that has been characteristically reported in the literature by multiple authors. ADC, apparent diffusion coefficient; EDAS, encephaloduroarteriosynangiosis. Reprinted with permission from Surmak et al. [27], 2020, *Journal of Alzheimer’s Disease*.

**Figure 5 ijms-25-03892-f005:**
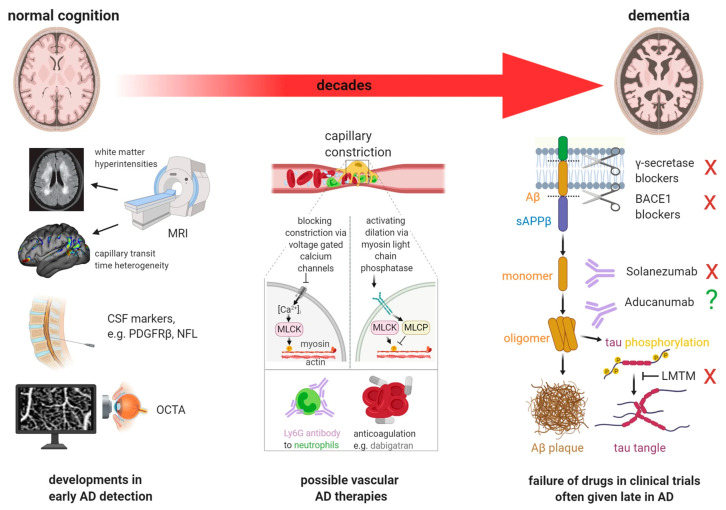
Composite figure showing interventions to diagnose and reduce cognitive decline at different stages of the transition from normal cognition to dementia in AD. The right third of the figure demonstrates that most clinical trials are initiated at relatively late stages of the disease, when cognitive decline is already apparent and irreversible synapse or neuron loss may have taken place. This may explain why drugs that block the γ or β secretases, antibodies to different forms of Aβ, and a drug that blocks tau aggregation have all failed (red crosses) to stop cognitive decline in AD. The left third of the figure illustrates emerging diagnostic approaches for early detection of AD, including MRI assessment of white matter hyperintensities and capillary transit time heterogeneity, assessment of biomarkers in the CSF, and non-invasive capillary imaging in the retina using, for example, optical coherence tomography angiography (OCTA). The middle third of the figure shows potential therapies to prevent or reverse the cerebrospinal flow decrease arising when Ca2+ activates myosin light chain kinase (MLCK) to evoke pericyte-mediated capillary constriction. These include blocking pericyte voltage-gated calcium channels to block Ca2+-evoked constriction, raising pericyte cGMP level (by activating guanylate cyclase receptors, blue membrane protein) to stimulate myosin light chain phosphate (MLCP) and thus evoke dilation, disrupting neutrophil surface interactions with endothelial cells or other cells using antibodies, or blocking thrombus formation with dabigatran. Reproduced from Korte et al. [9], 2023, an open-access article under the CC BY license.

**Figure 7 ijms-25-03892-f007:**
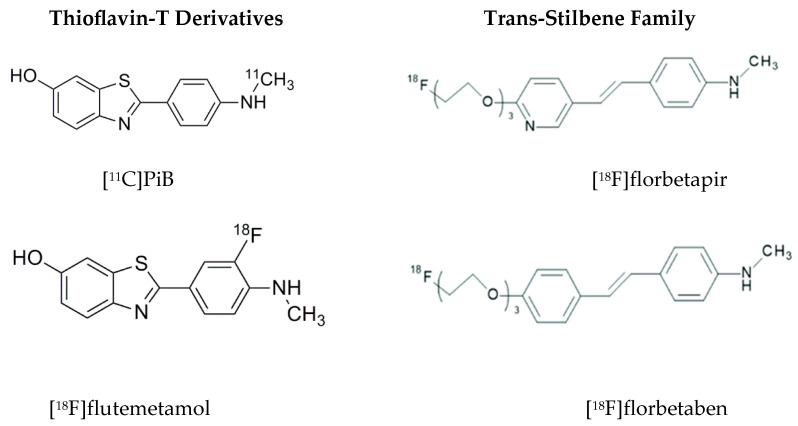
Chemical structure of FDA-approved amyloid-PET imaging probes from two distinctive structural families.

**Table 1 ijms-25-03892-t001:** Main issues and questions raised in this article.

Topic	Issue	Question	Common Perception	Independent Evidence
Cause of AD	Cortical amyloid deposition	Valid?	Yes	No
Diagnosis	ATN classification	Valid?	Yes	No
Amyloid positivity	By PET	Valid marker of AD?	Yes	No
Amyloid positivity	By CSF/plasma tests	Valid marker of AD?	Yes	No
Passive immunotherapy	Efficacy	Effective?	Yes	No
Passive immunotherapy	Amyloid removal	Therapy-induced?	Yes	No
Passive immunotherapy	ARIAs	Therapy-induced?	Yes	Yes
Passive immunotherapy	Brain volume loss	Therapy-induced?	Unknown	Yes

## Data Availability

Data are contained within the article.

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
