# Peer review of "Alzheimer’s Amyloid Hypothesis and Antibody Therapy: Melting Glaciers?"

_ijms, 2024, doi:10.3390/ijms25073892_

Round 1
Reviewer 1 Report
Comments and Suggestions for Authors
Reviewer comments
Alzheimer's disease is the most common neurodegenerative disease and cases of AD are continuously increasing throughout the globe. There are many therapies available; however, the success rate of all is not up to the mark. In this paper, the authors have discussed the amyloid hypothesis and the current scenario and success rate of the anti-amyloid immunotherapy for Alzheimer's disease. The article has briefly discussed the clinical efficacy of FDA-approved monoclonal anti-amyloid therapy. Further, the article highlights the FDA-approved amyloid PET imaging probes as a diagnostic tool. Overall article has emphasized more practical observation of the negative impact of this present immunotherapy on the brain.
Recommendation
1. The paper is scientifically sound, and written in an organized manner.
2. Authors have raised worthy questions about AD therapeutics and their target.
3. References are appropriate.
Scientific suggestion/s
1. Give a summary table for all the questions raised in the article. A table should have the names of therapeutic strategies, their major drawback, and how to overcome these drawbacks.
2. Provide a schematic representation of the pathophysiology of AD as discussed in the article and with the acceptable/effective therapeutic target/s and diagnostic biomarkers.
3. According to the authors what would be the future of mAb therapeutics in AD and how it could be improved?
4. What is the best replacement for anti-amyloid therapy?
Reviewer 2 Report
Comments and Suggestions for Authors
1. Section 2 is long. Please shorten the section and focus on Amyloid beta in AD pathology.
2. Please present the structures of AD treatments (e.g., aducanumab, lecanemab, donanemab, etc.) and PET imaging agtents (e.g., florbetaben, florbetapir, flutemetamol, PIB, etc.) along with their chemical/biological properteis.
3. Please provide the full name of each abbreviated terms in manuscript.
4. Please shorten some long sentences.
5. As a review paper, the authors should provide future directions of the research field. However, it is not clearly presented in the manuscript. Please add discussion of this aspect in conclusions.
6. Please double check the information of references. Some information (e.g., volume or page number) of references is missing.
Comments on the Quality of English LanguageSome sentences are too long. Please shorten those sentences (longer than 6 lines).
